# Secuer: Ultrafast, scalable and accurate clustering of single-cell RNA-seq data

**Nana Wei**[1], **Yating Nie**[1], **Lin Liu**[2]*, **Xiaoqi Zheng**[3]*, **Hua-Jun Wu**[4]*

**1** Department of Mathematics, Shanghai Normal University, Shanghai, China, **2** Institute of Natural Sciences, MOE-LSC, School of Mathematical Sciences, CMA-Shanghai, SJTU-Yale Joint Center for Biostatistics and Data Science, Shanghai Jiao Tong University and Shanghai Artificial Intelligence Laboratory, Shanghai, China, **3** Center for Single-Cell Omics, School of Public Health, Shanghai Jiao Tong University School of Medicine, Shanghai, China, **4** Center for Precision Medicine Multi-Omics Research, School of Basic Medical Sciences, Peking University Health Science Center and Peking University Cancer Hospital and Institute, Beijing, China

* linliu@sjtu.edu.cn (LL); xqzheng@shsmu.edu.cn (XZ); hjwu@pku.edu.cn (HJW).

## Abstract

Identifying cell clusters is a critical step for single-cell transcriptomics study. Despite the numerous clustering tools developed recently, the rapid growth of scRNA-seq volumes prompts for a more (computationally) efficient clustering method. Here, we introduce Secuer, a Scalable and Efficient speCtral clUstERing algorithm for scRNA-seq data. By employing an anchor-based bipartite graph representation algorithm, Secuer enjoys reduced runtime and memory usage over one order of magnitude for datasets with more than 1 million cells. Meanwhile, Secuer also achieves better or comparable accuracy than competing methods in small and moderate benchmark datasets. Furthermore, we showcase that Secuer can also serve as a building block for a new consensus clustering method, Secuer-consensus, which again improves the runtime and scalability of state-of-the-art consensus clustering methods while also maintaining the accuracy. Overall, Secuer is a versatile, accurate, and scalable clustering framework suitable for small to ultra-large single-cell clustering tasks.

**Data Availability Statement:** The datasets in this study are all publicly available: Biase, Yan, Goolam, Deng, Pollen and Kolodziejczyk (https://hemberg-lab.github.io/scRNA.seq.datasets/), Worm neuron, 10X PBMC, CITE PBMC, and Mouse retain and Human kidney (https://github.com/ttgump/scDCC/

## Author summary

Recently, single-cell RNA sequencing (scRNA-seq) has enabled profiling of thousands to millions of cells, spurring the development of efficient clustering algorithms for large or ultra-large datasets. In this work, we developed an ultrafast clustering method, Secuer, for small to ultra-large scRNA-seq data. Using simulation and real datasets, we demonstrated that Secuer yields high accuracy, while saving runtime and memory usage by orders of magnitude, and that it can be efficiently scaled up to ultra-large datasets. Additionally, with Secuer as a subroutine, we proposed Secuer-consensus, a consensus clustering algorithm. Our results show that Secuer-consensus performs better in terms of clustering accuracy and runtime.

tree/master/data), TAM FACS (https://figshare.com/projects/Tabula_Muris_Senis/64982), MCA (https://figshare.com/articles/dataset/MCA_DGE_Data/5435866), and COVID19 (GSE158055, https://www.ncbi.nlm.nih.gov/geo/query/acc.cgi?acc=GSE158055). Python implementation of Secuer is available on GitHub (https://github.com/nanawei11/Secuer).

**Funding:** This work was supported by National Key R&D Program of China (2018YFA0900600 to X.Z.); the Fundamental Research Funds for the Central Universities (PKU2022LCXQ027 - Clinical Medicine Plus X - Young Scholars Project, Peking University and BMU2021YJ064 to H.J.W. and Shanghai Jiao Tong University Start-up Grant WF220441912 to L. L.); National Natural Science Foundation of China (61572327 and 61972257 to X.Z., 12090024 and 12101397 to L.L., 32270683 to H.J.W.); Natural Science Foundation of Shanghai (20JC1413800 to X.Z., 21ZR1431000 and 21JC1402900 to L.L.); Shanghai Municipal Science and Technology Major Project (2021SHZDZX0102 to L.L.); Pujiang National Lab Grant (P22KN00524 to L.L.). The funders had no role in study design, data collection and analysis, decision to publish, or preparation of the manuscript.

**Competing interests:** The authors have declared that no competing interests exist.

This is a *PLOS Computational Biology* Methods paper.

## Introduction

In the past decade, single-cell RNA sequencing (scRNA-seq) has transformed our understanding of development and disease through profiling the whole transcriptome at the cellular level [1,2]. It has been widely used to unravel cell-to-cell heterogeneity and gain new biological insights, owing to its ability to identify and characterize cell types in complex tissues [3]. Unsupervised clustering approaches have played a central role in determining cell types. However, the scale of scRNA-seq experiments has been rapidly climbing in recent years, amounting to several datasets profiling over 1 million cells [4–6]. The increasing sample size renders many of the existing scRNA-seq clustering algorithms obsolete and prompts for developing a new generation of clustering algorithms that are efficient and scalable to large (500,000 ~ 5 million cells) or even ultra-large (> 5 million cells) scRNA-seq datasets.

Currently, two clustering algorithms, Louvain and Leiden, prevail in scRNA-seq analysis and have recently been implemented in numerous tools such as Seurat [7] and Scanpy [8]. Both algorithms aim to partition a graph into connected subgraphs by iteratively aggregating nodes: Louvain infers clusters by maximizing modularity [9], and Leiden is a variant of Louvain by using a local node-moving technique [10]. However, both algorithms are not well scaled to ultra-large datasets. For instance, for a dataset consisting of 10 million cells, Louvain and Leiden usually take 45 minutes and more than 1 hour for clustering, and both unreliably and unreasonably overestimate the number of clusters (as many as 1 million, see details in the Results section).

To address this gap, we present Secuer, a superfast and scalable clustering algorithm for (ultra-)large scRNA-seq data analysis based on spectral clustering. Spectral clustering has been one of the most popular clustering techniques due to its ease of use and flexibility of handling data with complicated shape or distribution [11], but with the caveat of high computational cost. We tailor the conventional spectral clustering to large scRNA-seq data based on an idea of representative/landmark selection in U-SPEC [12–14], leveraging the following three key elements in Secuer: First, we pivot $p$ anchors from all $N$ cells ($p \ll N$) and construct a weighted bipartite graph between cells and anchors by a modified approximate $k$-nearest neighbor (MAKNN) algorithm, which greatly accelerates the runtime of our method. Second, we determine the weights of the bipartite graph by a locally scaled Gaussian kernel function to capture the local geometry of the cell-to-anchor similarity network, which improves the accuracy of our method. Third, we design two optional approaches to automatically infer the number of clusters—$K$, which avoids manually choosing some $K$ by users.

We evaluate Secuer against three extensively used methods for scRNA-seq clustering including Louvain, Leiden, and k-means, using 31 simulated datasets with the number of cells ranging from 10,000 to 40 million. Secuer utilizes much shorter runtime than existing methods without deteriorating the clustering accuracy. In particular, Secuer is 5 times faster than k-means and 12 times faster than Louvain/Leiden for ultra-large datasets. Moreover, Secuer infers the number of clusters in the anchor space. The cluster number estimates by Secuer are still accurate when the sample size is larger than 5 million, in which case both Louvain and Leiden fail to produce any reasonable estimates. We then evaluate all the aforementioned methods in 15 real datasets with the number of cells ranging from 49 to 1.46 million (S1 Table), and find that Secuer yields more or comparably accurate clustering results than the other methods, and saves 90% of runtime in general.

With Secuer as a subroutine, we also develop a consensus clustering method, Secuer-consensus, by aggregating multiple clustering results obtained by Secuer to further boost clustering accuracy and stability. Compared to the popular consensus clustering algorithm SC3 [15], Secuer-consensus attains better clustering accuracy on 14 benchmark datasets, is in general 100 times faster, and can work on large datasets in which SC3 even fails to produce any useful output. Compared to Specter [16], another consensus clustering method for large-scale scRNA-seq data published recently, Secuer-consensus shows superior performance in both accuracy and speed on large datasets. In summary, Secuer and Secuer-consensus are accurate and scalable algorithms that provide a general framework for efficient (consensus) clustering of small to ultra-large scRNA-seq datasets that are being produced by a growing array of single cell transcriptomic projects.

## Results

### Overview of Secuer

The workflow of Secuer is illustrated in Fig 1. To improve computational efficiency, Secuer starts by randomly sampling a subset of $p'$ (10,000 by default) cells from all $N$ cells (Fig 1A and 1B). $p$ anchors are then obtained, which are the centroids of the identified clusters by applying k-means to the above random subsample (Fig 1B). Next the $k$-nearest anchors of all cells are determined by the MAKNN algorithm (Fig 1C). Then, a weighted bipartite graph between anchors and cells is constructed, with similarities between anchors and cells quantified by a locally scaled Gaussian kernel that the bandwidth parameter of each cell is defined as the

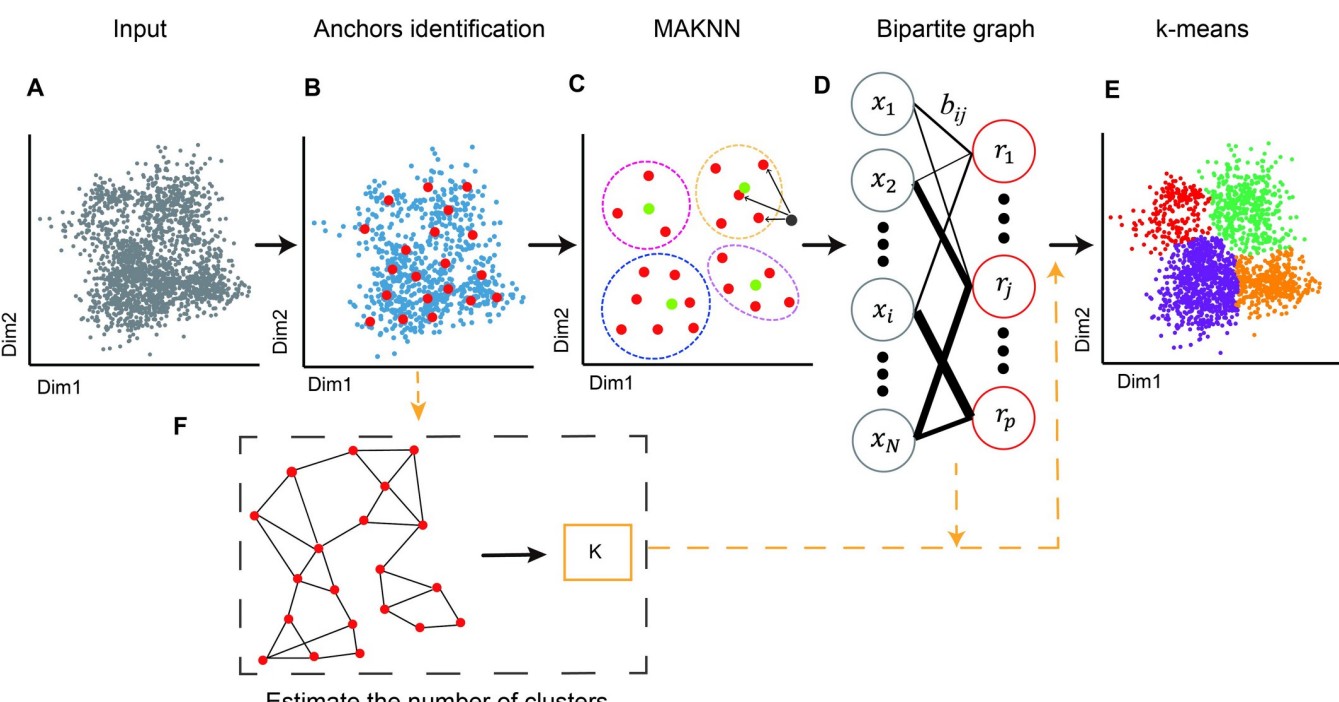

**Fig 1. Overview of the Secuer algorithm.** (A) Secuer takes the matrix in which rows are cells and columns are features as input. (B) Secuer obtains $p$ anchors (red points) by using k-means on a random subset of $p'$ cells from all $N$ cells (blue points). (C) The MAKNN algorithm step aims to find the $k$ nearest anchors for each cell (green points). (D) A weighted bipartite graph is constructed, with nodes representing cells (donated as $x$) and anchors (donated as $r$) and weights computed by a locally scaled Gaussian kernel distance. (E) Secuer applies k-means to the eigenvectors of the graph Laplacian of the weighted bipartite graph to obtain the final clustering results. (F) Secuer estimates the number of clusters based on the graph of the anchors or based on the eigenvalues of the graph Laplacian of the weighted bipartite graph of cells and anchors.

average distance between the cell and its *k* nearest neighbor anchors, to better capture the local geometry of the gene expression landscape (Fig 1D). Finally, we compute the first *K* eigenvectors of the graph Laplacian by transfer cuts (T-cut) algorithm [17] and obtain the final clustering results by off-the-shelf clustering algorithms such as k-means (by default) (Fig 1E). Secuer also automatically infers the number of clusters by either applying a community detection-based technique on the graph of the anchors (by default) (Fig 1F), or using the number of near-zero eigenvalues of the graph Laplacian (see Materials and Methods) [11].

We observed that the locally scaled Gaussian kernel is better suited to model certain scRNA-seq data than the non-locally scaled Gaussian kernel used in U-SPEC [12] (S1 Fig). In addition, we studied how two important tuning parameters– *p* (the number of anchors) and *k* (the number of nearest neighbors in MAKNN)–affected the clustering results in Secuer, and found that *p* = 1000 and *k* = 7 produced superior results (S2 Fig), which are therefore recommended as the default values when implementing Secuer.

## Secuer performance on simulated datasets

Clustering ultra-large scRNA-seq datasets is computationally intensive in terms of both runtime and memory usage. We generated a series of scRNA-seq datasets with an increasing number of cells ranging from 10,000 to 40 million (see Materials and Methods) to test the performance of Secuer and three widely used clustering methods: k-means, Louvain and Leiden. The number of clusters is determined by the default parameters except for k-means. For the ease of comparison, the reference/ground-truth (see Materials and Methods) number of clusters are given to k-means as input. The clustering accuracy is measured by the Adjusted Rand Index (ARI) [18] and the Normalized Mutual Information (NMI) [19].

First, we compared the runtimes of different methods. Secuer is the fastest on large and ultra-large datasets, in particular for ultra-large datasets, where Secuer is 5 times faster than k-means, and 12 times faster than Louvain/Leiden. Though k-means is faster for small datasets, the runtime only differs slightly from Secuer (below 10 seconds) (Fig 2A). In contrast, Louvain and Leiden are much slower than Secuer and k-means when applied to datasets over all scales, and fail to process the datasets of more than 10 million cells. Note the runtime of Louvain and Leiden does not monotonically increase with the sample size, possibly due to the much higher number of estimated clusters for larger datasets (Fig 2C). We then investigated the memory

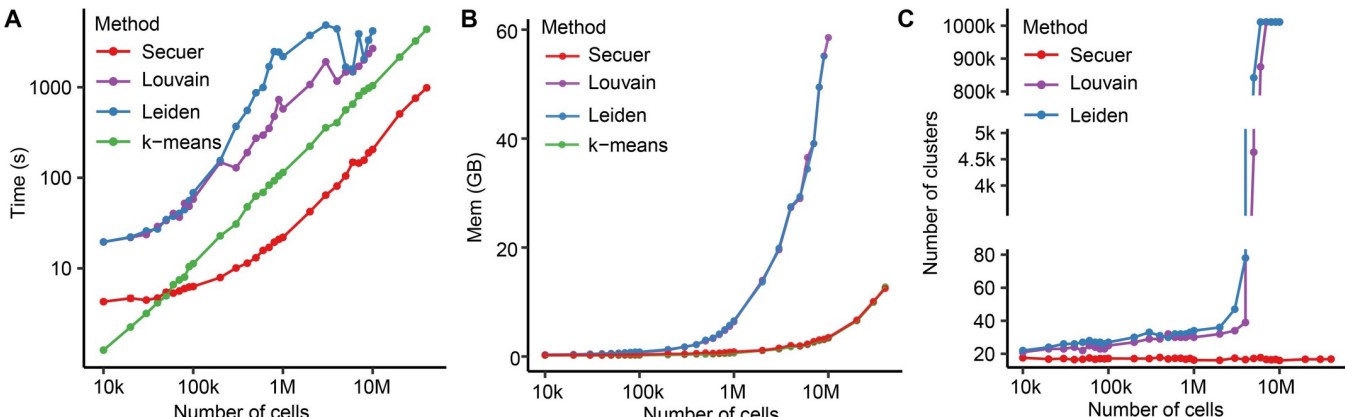

**Fig 2. The performance of different methods on the simulated datasets.** (A) The clustering runtime vs. the number of cells in the simulated datasets for all four methods. (B) The memory usage vs. the number of cells in the simulated datasets for all four methods. (C) The estimated number of clusters vs. the number of cells in the simulated datasets for three out of four methods: Secuer, Louvain and Leiden. k: thousand, M: million.

usage of different methods. We observed that Secuer and k-means consume the least amount of memory in all datasets. In comparison to Louvain and Leiden, Secuer only requires one-tenth of their memory usage when the sample size is over 1 million (Fig 2B). We also evaluated the clustering accuracy of all the methods and found that they all performed similarly when the sample sizes are less than 5 million. However, Louvain and Leiden performed poorly when there are more than 5 million cells, suggesting that they should not be applied directly to ultra-large datasets, even without concerning the computational cost (S3A Fig).

Next, we investigated the accuracy of Secuer in inferring the number of clusters (*K*). We set the true *K* = 19 across all simulated datasets over different sample sizes (see Materials and Methods). Fig 2C displays the number of clusters identified by Secuer, Louvain and Leiden, with sample sizes varying from 10,000 to 40 million. The numbers of clusters estimated by Secuer only fluctuate mildly around 18 across different simulations. However, neither Louvain nor Leiden can infer *K* correctly in ultra-large datasets when the number of cells is over 5 million (Fig 2C). Notably, the estimated numbers of clusters by Louvain and Leiden are enormously upwardly biased, and such a bias cannot be easily fixed by simply lowering the resolution parameter to even 0.0001 (S3B Fig). This further suggests that Secuer should be favored over Louvain and Leiden when analyzing large to ultra-large datasets.

Finally, we compared Secuer with the vanilla spectral clustering algorithms (VSC), especially in runtime. To this end, we divided the clustering procedure into three steps: constructing weighted bipartite graph, solving the eigen-problem, and the final clustering. We investigated the runtime of each step between Secuer and VSC. Secuer has significantly reduced runtime on the graph construction step in larger datasets and the eigen-solving step in datasets over all scales, due to the use of the T-cut algorithm on the anchor-based bipartite graph (S3C and S3D Fig, see Materials and Methods).

## Secuer performance on large real datasets

To evaluate the performance of our method on large real datasets, we collected three scRNA-seq datasets recently reported: COVID19 dataset profiling 1.46 million immune cells isolated from 196 patients; MCA dataset containing 325,486 cells from multiple major mouse organs; and Mouse brain dataset consisting of over 1 million cells from two E18 mice. For each dataset, we performed all clustering methods 10 times and reported the average runtimes and clustering accuracy.

Secuer has the shortest runtime on all datasets (Fig 3A). In particular, for the COVID19 dataset with more than 1 million cells, the runtime of Secuer is < 1 minute on average, which is 3 times faster than k-means, and 24 times faster than Louvain and Leiden. In the meanwhile, Secuer achieves competitive accuracy in general compared to other methods measured by both ARI (Fig 3B) and NMI (S4A Fig). An example of the clustering results by all methods are displayed in Uniform Manifold Approximation and Projection (UMAP) [20] visualization (Fig 3C–3I), in which the cell type labels are obtained from Xie et al. [21]. Under default parameters, Secuer exhibits more apparent cell type separations than Louvain and Leiden, both of which obtain 1.5 times more clusters showing ambiguous patterns, especially on the right region of the plot with a lot of small clusters overlaying with each other (Fig 3D–3F). The performance of Louvain and Leiden is improved when the number of clusters is tuned to be the same as that of Secuer. However, by doing this they fail to distinguish the Interneurons (cluster 1 in Secuer) and Neural stem/precursor cells (cluster 6 in Secuer) in the lower left corner of the plot (Fig 3G and 3H). Likewise, k-means also fail to distinguish the two clusters even though the original number of cell type labels is given (Fig 3I). Taken together, these findings demonstrate that Secuer is an efficient and accurate algorithm for clustering large scRNA-seq datasets.

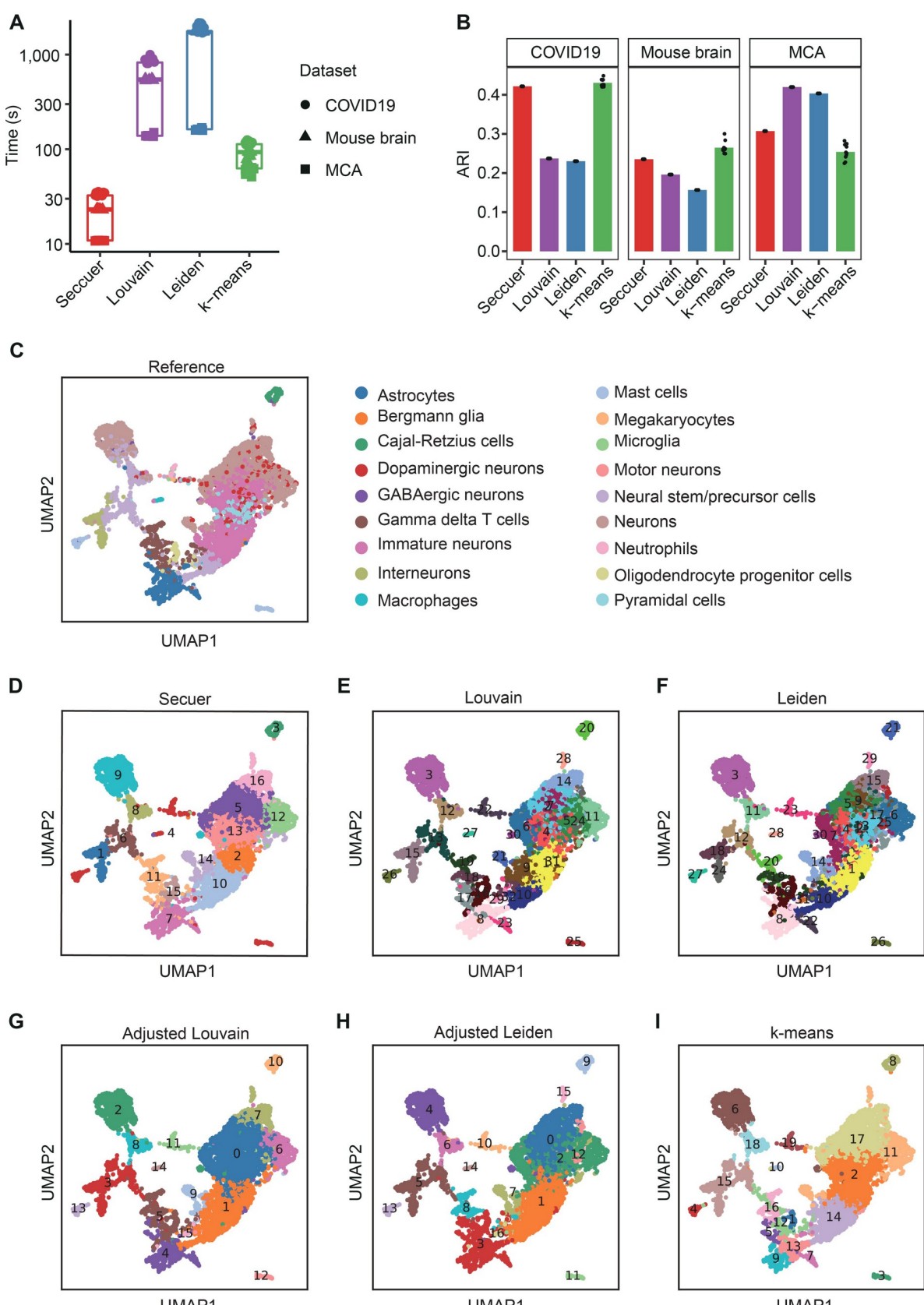

**Fig 3. The performance of different methods on large real datasets.** (A) The clustering time of different methods. (B) The ARI of different methods on three large datasets. (C-I) UMAP visualization of the Mouse brain dataset for the different methods. Reference (C) illustrates the ground-truth cell type labels obtained from the original study. Secuer (D), Louvain (E), and Leiden (F) display clustering results by using their default parameters. Adjusted Louvain (G) and adjusted Leiden (H) refer to the clustering results by setting the resolution parameter to 0.3. k-means (I) represents the clustering results given the ground-truth number of clusters in (C).

## Secuer performance on well-annotated benchmark datasets

To further investigate the performance of our method in small and moderate benchmark datasets, we applied the same analyses as above on six gold-standard and six-silver standard scRNA-seq datasets introduced by SC3 [15], scDCC [22], and MARS [4]. These datasets, with numbers of cells varying from 49 to 110,832, are widely used to benchmark the performance of new clustering methods because the cell type labels with high confidence are available.

As expected, Secuer and k-means remain to have the shortest runtimes, which are at least 10 times faster than Louvain and Leiden across all 12 datasets (Fig 4A). In terms of accuracy, Secuer outperforms Louvain and Leiden on 7 out of 12 datasets (mean ARI difference > 0.05), and is substantially better on 4 out of 12 (Goolam, Biase, Human kidney and Mouse retina, mean ARI difference > 0.2) (Fig 4B and 4C). The distribution of ARI across all datasets demonstrates that Secuer is highly competitive even only in terms of clustering accuracy (Fig 4D). Likewise, similar trends are observed when measuring accuracy using NMI (S4B–S4D Fig). k-means sometimes achieves better accuracy than the other methods, but the clustering results are exceedingly inconsistent across different initializations. For instance, the range of ARI in different runs is 0.5 in the Biase dataset (Fig 4B). Furthermore, we found that k-means is sensitive to changes in the data preprocessing pipelines. In most cases, the mean NMIs of k-means with and without preprocessing step differ by a large margin, especially for Goolam and CITE PBMC datasets (mean NMI difference > 0.5, S5 Fig). Using the Mouse retina dataset as an example, The UMAP visualization suggests that the clusters inferred by Secuer are more aligned with the given reference cell-type annotations (Fig 4E–4I), regarded as the ground truth. Notably, cluster 1 identified by Secuer is perfectly matched with the reference cluster 5, but none of the other approaches are capable of recovering this reference cluster. Finally, we examined the accuracy of the estimated number of clusters by different methods on all 12 datasets and discovered that Secuer outperformed Louvain and Leiden (S6 Fig). These results demonstrate that Secuer is also competitive in terms of clustering accuracy for small and moderate scRNA-seq datasets.

## Secuer-consensus performance on fourteen benchmark datasets

Most of the unsupervised clustering algorithms may yield inconsistent results due to random initialization and different parameter settings. To resolve this common problem, consensus clustering aggregates multiple outputs generated by different clustering algorithms or by the same algorithm but with varied parameter settings, to produce consensus clusters that are expected to be more stable and accurate [15,23]. Just as in the case of usual clustering, consensus clustering method applicable to large-scale scRNA-seq data is also lacking. To bridge this gap, with Secuer as a subroutine, we developed Secuer-consensus, a highly efficient consensus clustering algorithm that ensembles multiple outputs of Secuer by fully taking advantage of its computational efficiency. The implementation of Secuer-consensus constitutes three steps (Fig 5A): First, Secuer is run for $M$ times with different distance metrics, including Euclidean and cosine distances, and different numbers of clusters estimated by different parameter settings to generate multiple clustering outputs. Second, an unweighted bipartite graph is constructed from the multiple outputs. Third, k-means clustering is performed on the unweighted bipartite graph (see Materials and Methods).

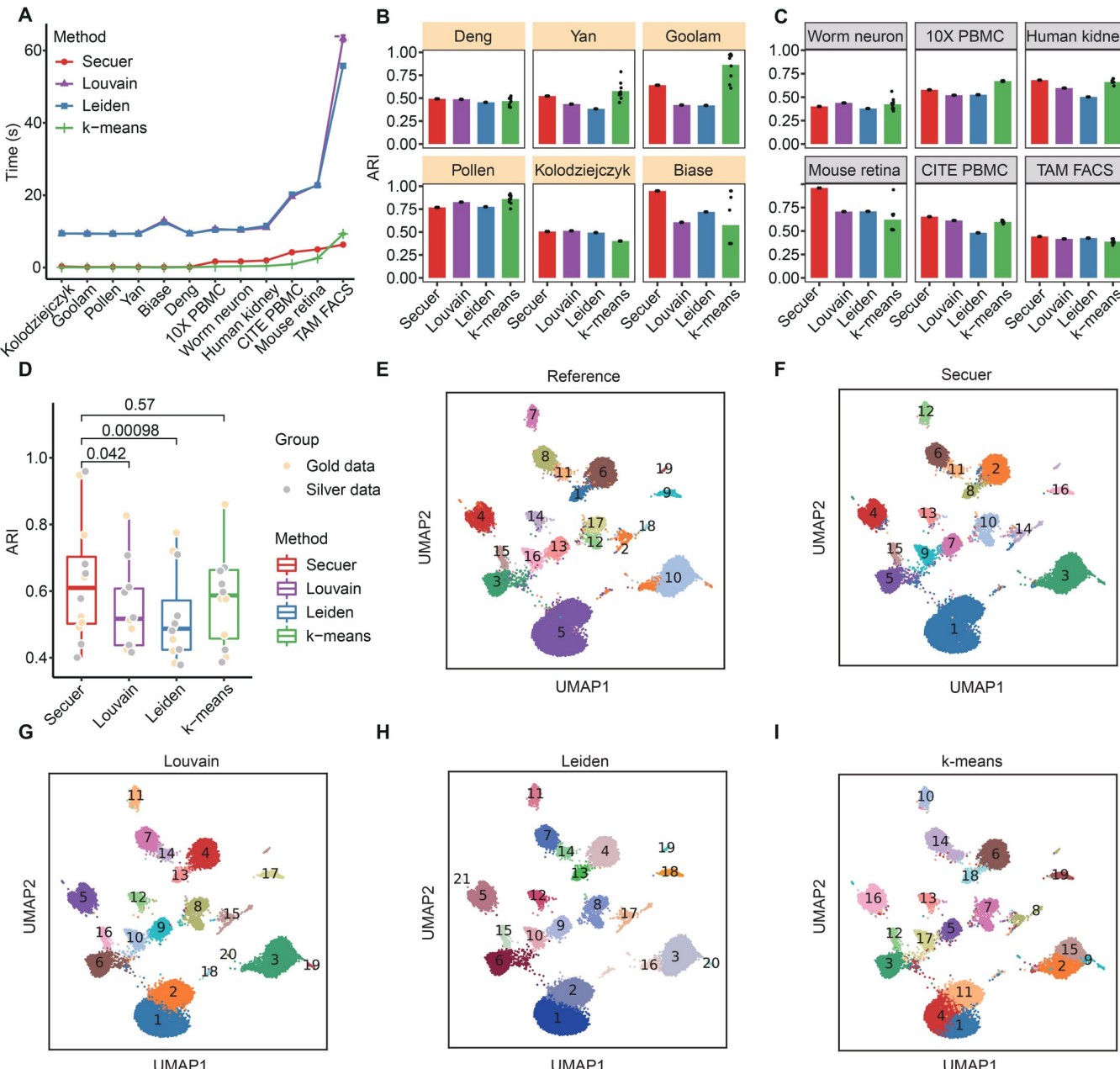

**Fig 4. Performance of Secuer on twelve gold and silver standard datasets.** (A) The clustering runtime of each method on all twelve datasets. (B-C) Accuracy of different methods, including k-means, Louvain, Leiden, and Secuer, on gold (B) and silver (C) standard datasets. (D) A boxplot showing the distribution of ARI of different methods on all datasets. (E-I) UMAP visualization of the ground-true cell type labels obtained from the original study, termed as reference (E) and clustering results from four different methods (F-I) on Mouse retina dataset.

We evaluated the performance of Secuer-consensus on 14 datasets, including 12 gold/silver standard datasets and 2 million scale datasets, by comparing it to the original Secuer and two consensus clustering methods SC3 and Specter. Specter requires the number of clusters as input, which is provided by our method. We first studied how many clustering outputs of Secuer ($M$) should be fed into Secuer-consensus and found that setting $M$ between 5 and 10 has generally better performance (S7 Fig). Hence, we set $M = 5$ for all datasets hereafter. It can be found that Secuer-consensus achieved similar runtime as Specter that is over 100 times

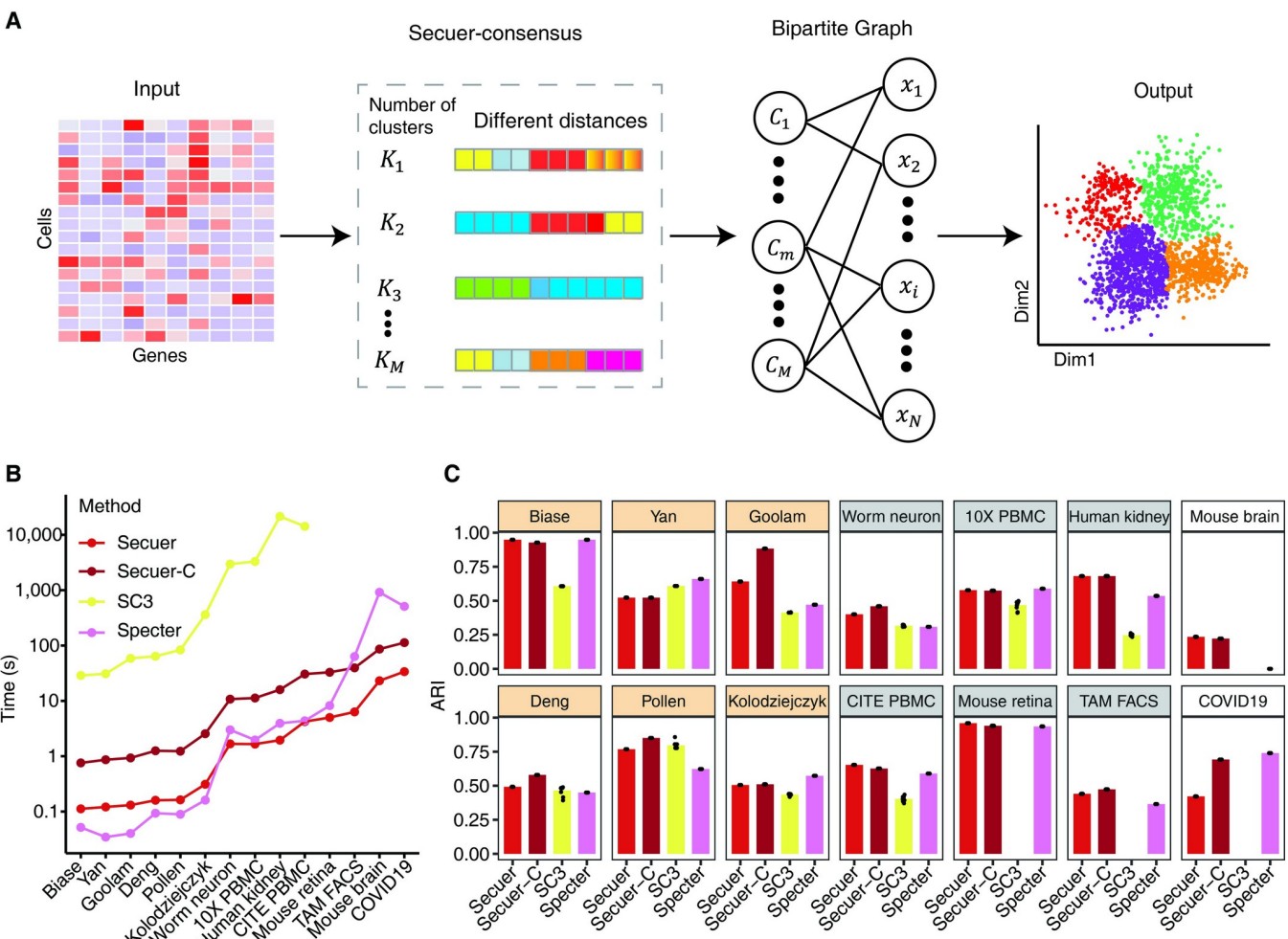

**Fig 5. Overview of the Secuer-consensus algorithm and the performance on fourteen scRNA-seq datasets.** (A) Secuer-consensus takes a matrix as input, with genes as the columns and cells as the rows, executes Secuer $M$ times to acquire multiple clustering outputs, and constructs an unweighted bipartite graph, with two sets of nodes respectively representing the clusters (denoted as $C$) and cells (denoted as $x$). Finally, k-means clustering is used to obtain a consensus grouping. (B) The clustering runtime for different methods. Secuer-C: short for Secuer-consensus. (C) The ARI for four methods on 14 benchmark datasets.

faster than SC3 in small and moderate datasets (Fig 5B). Despite suboptimal compared with Specter on gold and some silver standard datasets, the runtime difference between Secuer-consensus and Specter is less than 8 seconds on average. However, when sample size increases, Secuer-consensus shows superior speed than Specter. In the case of processing million scale single cell datasets, Secuer-consensus takes 86 seconds for Mouse dataset ($N$ = 1,011,462) and 112 seconds for COVID19 dataset ($N$ = 1,462,702), while Specter takes 15 minutes and 8.5 minutes, respectively. Furthermore, Secuer-consensus can be accelerated by using parallel computation: for example, the average runtime is decreased by 50% on datasets with more than 1 million cells using 3 cores (S8 Fig).

Regarding accuracy, Secuer-consensus is approaching or superior to Specter and SC3, depending on the datasets. Specifically, Secuer-consensus surpasses SC3 on 9 out of 10 datasets (SC3 failed to process the four larger datasets in our settings) and Specter on 7 out of 14 (mean ARI difference > 0.05), while substantially outperforms SC3 on four datasets (Biase, Goolam, Human kidney and CITE PBMC datasets, ARI difference > 0.2) and Specter on two datasets (Goolam and Mouse datasets, ARI difference > 0.2) (Fig 5C). These findings indicate that

Secuer-consensus is an appealing option for consensus clustering small to large scRNA-seq datasets.

## Discussion

Identifying cell clusters is a critical step for scRNA-seq data analysis. Computationally efficient and scalable methods are urgently needed due to the rapidly expanding volume of scRNA-seq data. In this work, we presented Secuer, a computationally efficient, ultra-scalable and accurate method for unsupervised clustering of scRNA-seq data. Secuer is on average 10 times faster than Louvain and Leiden while exhibiting similar accuracy in 15 benchmark datasets, covering different sequencing technologies, with the number of cells ranging from 49 to 1.4 million. Secuer can efficiently scale to ultra-large scRNA-seq datasets of more than 10 million cells, when neither Louvain nor Leiden is even able to process the data. For instance, Secuer can cluster a scRNA-seq dataset of 10 million cells within 3 minutes, which is 6 times faster than k-means, one of the most efficient off-the-shelf clustering algorithms. In addition, Secuer can reliably estimate the number of clusters regardless of the number of cells in the data, whereas both Louvain and Leiden usually erroneously identify > 0.9 million clusters for datasets with over 5 million cells.

With Secuer as a subroutine, we also proposed a consensus clustering method, Secuer-consensus, by aggregating multiple Secuer runs with an array of different parameter settings. Secuer-consensus surpasses or approaches Secuer, SC3 and Specter in all 14 benchmark datasets. Notably, Secuer-consensus only takes less than 1% the runtime of SC3 for these datasets, and 15% the runtime of Specter on the million datasets. By parallelizing the computation, our method can further reduce runtime on large datasets. This allows Secuer-consensus to cluster cell types with improved stability and accuracy compared to the non-consensus-based methods, while being substantially more efficient than competitive consensus clustering methods.

Overall, our new clustering framework strikes a good balance between accuracy, computational cost and scalability. It is an appealing choice for clustering large-scale scRNA-seq atlas, and can also be easily incorporated into any online scRNA-seq computational platforms for real-time analysis. The computational efficiency of Secuer also makes it a building block for scalable consensus clustering, demonstrated by the superior performance of Secuer-consensus than other competing methods. Secuer is also flexible enough to be adapted to a wide range of clustering algorithms beyond spectral clustering, such as Louvain or upcoming new approaches, to enhance their efficiency and scalability. As the rapid development of droplet-based single cell technologies, we expect our framework can eventually be applied to identify cell clusters in large-scale omics data other than scRNA-seq, such as scATAC-seq, CyTOF and image-based spatial data.

## Materials and methods

### Spectral clustering

Spectral clustering is a popular clustering algorithm originated in spectral graph theory [24]. Given a graph $G = \{X, E, S\}$, where $X = \{x_1,\ldots,x_N\}$ is a set of $N$ data points and each $x_i$ is a $d$-dimensional vector, $E$ is a set of edges and $S = [S_{in}]_{i,n = 1,2,\ldots,N}$ is a weighted adjacency or similarity matrix, spectral clustering aims to divide the graph into connected subgraphs in which the within-group edge weights are maximized while the between-group edge weights are minimized. A typical spectral clustering algorithm constitutes the following main steps: 1) constructing an adjacency matrix $S$ for data points based on certain distances (e.g., Euclidean, cosine); 2) computing the graph Laplacian matrix, i.e., $L = D - S$, where $D$ is the degree matrix of the graph, a diagonal matrix with diagonal elements equal to the row sums of $S$; 3) calculating the eigenvectors corresponding to the $K$ smallest eigenvalues of the (normalized) Laplacian

and arranging them in a matrix $V$ by columns; 4) clustering the row-normalized $V$ into $K$ groups using conventional clustering algorithms, such as k-means or hierarchical clustering. Spectral clustering has several variants by using different forms of the Laplacian matrix [11]. Among them, Normalized cut (Ncut) is the most widely adopted method [25].

## Secuer

Our method is comprised of four steps: 1) identifying anchors; 2) estimating the number of clusters; 3) applying the MAKNN algorithm to construct the bipartite graph between anchors and cells; 4) partitioning the bipartite graph partitioning. Among them, the step of identifying anchors to generate an adjacency matrix of cell-by-anchor to replace the original dense similarity matrix is key for drastically improving the clustering runtime and memory usage. Given a gene expression matrix $X \in \mathbb{R}^{N \times d}$, where $N$ and $d$ are, respectively, the number of cells and the number of genes. The vanilla spectral clustering requires: 1) $O(N^2 d)$ time to build the adjacency matrix and 2) $O(N^3)$ time to solve the eigen-problem [13]. In contrast, Secuer reduces the time complexity to $O(Np^{\frac{1}{2}}d)$ in 1) by using the bipartite graph representation, and reduces the time complexity to $O(NK(K+k)+p^3)$ in 2) by using transfer-cuts (T-cut) [12,17] on the weighted bipartite graph to solve the eigen-problem, where $k$ is the number of neighboring anchors and $K$ is the estimated number of cell clusters.

## Identifying anchors

Given the gene expression matrix $X = \{x_i\}_{i=1,..,N}$, where $x_i \in \mathbb{R}^d$ represents the expression profile of one single cell, we selected $p$ (default by 1,000) anchors to bypass computing the original large and dense similarity matrix. The idea behind this step is to use a small set of landmark points to approximately represent the underlying geometry of the data. In detail, we first randomly selected $p'$ (default by $10p$) candidate cells from all $N$ cells such that $p<p'\ll N$, and then group candidate cells into $p$ clusters by k-means clustering. The final anchors are the centroids of the $p$ clusters, denoted as $r = \{r_1, r_2,. . .,r_p\}$. Note that, for datasets with cells less than 10,000, we applied k-means on the dataset directly to identify anchors.

## Estimating the number of clusters

The number of clusters is usually given *a priori* for vanilla spectral clustering algorithms. However, the true number of cell types is seldomly available in practice. In the current version of Secuer, we implemented two approaches for estimating the number of clusters. Inspired by community-detection algorithms that infer the number of clusters using a resolution parameter, we constructed a graph with only anchors as nodes and estimate the number of clusters by the community-detection-based approach used in Louvain algorithm, which is also used as the default option in Secuer.

Another option is to use the bipartite graph between anchors and cells. It is proven that the number of near-zero eigenvalues of the graph Laplacian matrix equals the number of the connected components of the underlying graph [11]. Inspired by this fact, we designed an approach consisting of the following five steps: 1) sorting the eigenvalues of the bipartite graph Laplacian matrix in ascending order; 2) dividing all the eigenvalues into 100 equal-sized bins $\mathcal{H} = \{(L_u, R_u)|u = 1, 2, \ldots, 100\}$ and count the number $C_u$ of eigenvalues falling into each bin; 3) computing the gap values $\Delta$ between consecutive bin pairs with $C_u>0$, denoted as $\Delta = \{L_{u'+1} - R_{u'}|u' : C_{u'} > 0\}$, and identify the $\alpha$-th greatest values $\Delta_\alpha$ in $\Delta$ (with default value set as $\alpha = 4$ by empirical experience from simulations and data analysis); 4) determining the bins with gap values greater than $\Delta_\alpha$ and obtain $u^* = \text{argmax}_{u'} \{R_{u'}|L_{u'+1} - R_{u'} > \Delta_\alpha\}$; 5) estimating the number of clusters by the number of eigenvalues falling into all bins with $u \leq u^*$.

## The MAKNN algorithm

Given $p$ anchors, we aimed to construct an $N{\times}p$ similarity matrix $S$ between all $N$ cells and $p$ anchors. However, this step can be computational expensive in terms of both runtime and memory usage for ultra-large datasets. To alleviate this problem, we used a modified approximate KNN algorithm to improve the computational efficiency. For large datasets, instead of building a large and dense adjacency matrix, the modified method aims to find $k$ nearest anchors approximately for each cell to build a sparse adjacency matrix of cell-to-anchor. Taking a cell $x_i$ as an example:

i.  All $p$ anchors are grouped into $o$ clusters using k-means, denoted as $\omega_1$, $\omega_2$,. . .,$\omega_o$.

ii.  Cell $x_i$ is then assigned to the closest cluster $\omega_l^{(i)}$ based on the Euclidean distance between cell $x_i$ and all cluster centers.

iii.  Find the nearest anchor of cell $x_i$ in $\omega_l^{(i)}$ denoted as $p^{(i)}$.

iv.  Apply KNN to $p$ anchors to obtain the $k'$ (default as $10{\times}k$) nearest neighbors of each anchor such that $k<k'$.

v.  Obtain the $k$ nearest anchors of cell $x_i$ based on the Euclidean distance between $x_i$ and the $k'$ nearest anchors of $p^{(i)}$.

In the above steps, we only need to calculate the $k'$ nearest neighbors of $p$ anchors. Then the neighbors of the anchor closest to the cell $x_i$ are treated as neighbors of that cell. Since $k{\ll}p{\ll}N$, the time complexity of this procedure is $O\left(podt + Nod + N\frac{p}{o}d + p^2d + Nkd\right) = O\left(Nod + N\frac{p}{o}d\right)$, which is minimized when $o$ is set to $p^{\frac{1}{2}}$ by equating $Nod$ and $N\frac{p}{o}d$, where $\frac{p}{o}$ is the average size of $o$ anchor clusters in step i) of the above algorithm and $t$ is the number of iterations of k-means. The above reasoning renders the order of the runtime $O\left(Np^{\frac{1}{2}}d\right)$.

We next used a locally scaled Gaussian kernel to measure the distance between cells and anchors, and obtain an $N{\times}p$ adjacency matrix $B$ with each row only keeping $k$ nonzero elements as

$$B = [b_{ij}], i = 1, 2, \ldots, N, j = 1, 2, \ldots, p,$$

$$b_{ij} = \begin{cases} \exp\left(-\dfrac{\|x_i - r_j\|^2}{2\sigma_i^2}\right), \text{if } r_j \in N_k(x_i), \\ 0, otherwise, \end{cases}$$

where $N_k(x_i)$ represents the $k$ nearest anchors of cell $x_i$, and $\sigma_i$ is the average distance between cell $x_i$ and its $k$ nearest anchors.

## Bipartite graph partitioning

Putting all cells and anchors together, we constructed a bipartite graph $G_b = \{X,r,W\}$, where $W$ is a $(N+p){\times}(N+p)$ weighted adjacency matrix, denoted as:

$$W = \begin{bmatrix} 0 & B \\ B' & 0 \end{bmatrix}.$$

Then spectral clustering is performed to partition the graph by solving the generalized eigen-problem:

$$Lv = \gamma Dv, \tag{1}$$

where $L = D - W$ is the Laplacian matrix, $D = \begin{bmatrix} D_X & 0 \\ O & D_p \end{bmatrix}$ is the degree matrix of the graph $G_b$.

For large datasets in which the number of cells is far greater than the number of anchors, $G_b$ is an unbalanced bipartite graph. We thus employed an efficient eigen-decomposition method T-cut [17], which turns the problem (1) into a computational eigenproblem:

$$L_p z = \lambda D_p z, \tag{2}$$

where $L_p = D_p - W_p$, $W_p = B' D_X^{-1} B$. Note that $D_p = diag(B' 1_N) = diag(W_p 1_p)$. Here $L_p$ is the Laplacian matrix of the graph $G_p = \{r, W_p\}$, $1_N = \underbrace{[1, 1, \ldots, 1]}_{N}'$ and $1_p = \underbrace{[1, 1, \ldots, 1]}_{p}'$.

Li et al. proved that the solution of the eigen-problem (1) on graph $G_b$ and the solution of the eigen-problem (2) on bipartite graph $G_b$ are equivalent [17]. Let $\{(\lambda_e, z_e)\}_{e=1}^{K}$ be the first $K$ eigenpairs of (2), where $0 = \lambda_1 < \lambda_2 < \cdots < \lambda_K < 1$, and $\{(\gamma_e, v_e)\}_{e=1}^{K}$ be the first $K$ eigenpairs of (1), where $0 \leq \gamma_e < 1$. According to Li et al., we have the following

$$\gamma_e(2 - \gamma_e) = \lambda_e,$$

$$v_e = \begin{bmatrix} \xi_e \\ z_e \end{bmatrix},$$

where $\xi_e = \frac{1}{1 - \gamma_e} P z_e$, and $P = D_X^{-1} B$ is the associated transition probability matrix from cells to anchors.

After normalizing the matrix $T = [\xi_1, \ldots, \xi_K]_{N \times K}$ to unit length, k-means is applied to the rows of normalized matrix to obtain the final clustering result. Note that k-means can be replaced by other clustering algorithms such as DBSCAN or hierarchical clustering. However, these methods are generally less efficient than k-means.

## Secuer-consensus

Taking advantage of the computational efficiency of Secuer, we proposed a consensus clustering method Secuer-consensus, which aggregates multiple clustering outputs from Secuer to boost the clustering stability and accuracy. The implementation is as follows. First, $M$ different base clustering results are obtained from Secuer, by varying the selection of anchors, the number of clusters, and the distance metrics between anchors and cells (Euclidean or cosine). Denote all the clusters in $M$ base clustering results as $C = \{C_1^1, \ldots, C_1^{K_1}, C_2^{K_1+1}, \ldots, C_2^{K_1+K_2}, \ldots, C_M^{K_C}\}$, where $K_C = \sum_{m=1}^{M} K_m$ and $K_m$ is the number of clusters in the $m$-th base clustering result and $C_m^e$ is the $e$-th clusters of $m$-th the base clustering result. Then, a bipartite graph $G_{XC} = \{X, C, \tilde{W}\}$ is constructed, in which the nodes are the cells and clusters and $\tilde{W}$ is an $(N+K_C) \times (N+K_C)$ adjacency matrix indicating whether a cell belongs to a cluster. An edge only appears between a cell and the clusters to which the cell belongs and no edges between different cells or between different clusters are allowed. $\tilde{W}$ can be written as:

$$\tilde{W} = \begin{bmatrix} 0 & \tilde{E} \\ \tilde{E}' & 0 \end{bmatrix},$$

$$\tilde{E}_{i,h} = \begin{cases} 1, x_i \in C_{\cdot}^h \\ 0, \text{otherwise} \end{cases},$$

where $\tilde{E}$ is an $N \times K_C$ matrix.

Similar to the Bipartite graph partitioning section, we next solved the eigenvalue of graph Laplacian of $G_{XC}$ by T-cut. That means solving the following eigenproblem:

$$\tilde{L}\tilde{\nu} = \tilde{\gamma}\tilde{D}\tilde{\nu}, \tag{3}$$

where $\tilde{L} = \tilde{D} - \tilde{W}$ is the Laplacian matrix, and $\tilde{D} = \begin{bmatrix} \tilde{D}_X & 0 \\ O & \tilde{D}_C \end{bmatrix}$ is the degree matrix of the bipartite graph $G_{XC}$.

The eigen-problem in (3) is equivalent to solving the following problem,

$$\tilde{L}_C\tilde{z} = \tilde{\lambda}\tilde{D}_C\tilde{z}, \tag{4}$$

where $\tilde{L}_C = \tilde{D}_C - \tilde{E}_C$ is the Laplacian matrix of $G_C = \{C, \tilde{E}_C\}$, $\tilde{E}_C = \tilde{E}'\tilde{D}_X^{-1}\tilde{E}$ is the adjacency matrix and $\tilde{D}_C = diag(\tilde{E}_C 1_{K_C})$ is the degree matrix.

Let $\{(\tilde{\lambda}_e, \tilde{z}_e)\}_{e=1}^K$ be the first $K$ eigenvalues and eigenvectors of (4), where $0 = \tilde{\lambda}_1 < \tilde{\lambda}_2 < \cdots < \tilde{\lambda}_K < 1$. Further let $\{(\tilde{\gamma}_e, \tilde{\nu}_e)\}_{e=1}^K$ be the first $K$ eigenpairs of (3), where $0 \le \tilde{\gamma}_e < 1$. Then we have:

$$\tilde{\gamma}_e(2 - \tilde{\gamma}_e) = \tilde{\lambda}_e,$$

$$\tilde{\nu}_e = \begin{bmatrix} \tilde{\xi}_e \\ \tilde{z}_e \end{bmatrix},$$

where $\tilde{\xi}_e = \frac{1}{1-\tilde{\gamma}_e}\tilde{P}\tilde{z}_e$, and $\tilde{P} = \tilde{D}_X^{-1}\tilde{E}$.

Then k-means is applied to the rows of normalized matrix $\tilde{T} = [\tilde{\xi}_1, \ldots, \tilde{\xi}_K]_{N \times K}$ to obtain the final clustering result.

## Benchmark datasets

Fifteen publicly available scRNA-seq datasets, including six gold-standard datasets, six-silver standard datasets and three ultra-large datasets, are used to evaluate the clustering accuracy of our method (see S1 Table for details) [1,5,26–40]. In six gold-standard datasets, cells are highly confident to be labeled as a specific cell type/stage according to their surface markers. In six silver standard datasets, the label of each cell is assigned by computational tools and manual annotation using *prior* knowledge by previous studies [31–33,36–38]. Although widely used to benchmark a newly proposed clustering method [4,15,22], these datasets are relatively small in size (containing from 49 to 110,832 cells per dataset), and thus insufficient to evaluate a method designed for much higher throughput considered in this paper. Thus, we also assessed our method using three large datasets [5,41], which consist of 1 million cells on average. The cell type labels in these datasets are collected from the original studies. In all the 15 datasets, the cell type labels are considered as ground-truth, also termed as "reference", throughout this study.

## Data preprocessing

The preprocessing involves four steps: 1) gene/cell filtering; 2) normalization; 3) selection of highly variable genes; 4) dimension reduction by PCA. The parameters of the preprocessing pipeline are the same for all datasets except for the gene/cell filtering step, which instead follows the criterions in the original studies. For six small gold standard datasets, we adopted a preprocessing strategy of gene filtering similar to that for SC3 [15]: genes are removed by the 'gene filter' function in the Scanpy package [8] if they are expressed in less than 10% or more than 90% of the cells. For larger datasets including 10X PBMC, Worm neuron, Human kidney, CITE PBMC and Mouse retain datasets, we filtered cells with fewer than one gene and retained genes expressed in at least one cell using Scanpy. For mouse brain, we excluded cells with fewer than 200 genes and mitochondrial genes with a UMI greater than 5%. Genes expressed in less than 3 cells were also removed. For TAM FACS, we retained genes expressed in at least 3 cells and cells with no less than 250 expressed genes and 5000 counts. For MCA, cells with fewer than 100 genes and genes with less than 3 cells were excluded. After filtering, raw count matrix was normalized and log-transformed to detect highly variable genes (HVG). Finally, Principal Component Analysis (PCA) was performed on the selected HVGs, and the top 50 PCs were retained for clustering. For COVID19, we downloaded the processed data provided by the authors. Scripts for all the above preprocessing steps are available at https://github.com/nanawei11/Secuer.

Note that two optional steps of data preprocessing, i.e., normalization and selection of HVGs, can potentially affect the clustering results [42,43]. Therefore, we compared the clustering accuracies of different methods both with and without these two steps (S5 Fig).

## Distance metrics

Denote $x_{ig}$, $i = 1,\ldots,N$, $g = 1,\ldots, d$ as the gene expression level of the cell $i$ in gene $g$. We used the following Euclidean and cosine distances as candidate distance metrics to build the bipartite graph between cells and anchors:

$$D_{Euclidean}(x_i, x_n) = \sqrt{\sum_{g=1}^{d}(x_{ig} - x_{ng})^2},$$

$$D_{cosine}(x_i, x_n) = 1 - \frac{\sum_{g=1}^{d} x_{ig} x_{ng}}{\sqrt{\sum_{g=1}^{d} x_{ig}^2}\sqrt{\sum_{g=1}^{d} x_{ng}^2}}.$$

## Clustering metrics

We compared the clustering accuracy of different methods using the ARI [14] and NMI [15], which are widely used indices for evaluating the clustering performance when the reference or the true cluster labels are known. ARI is defined as

$$ARI = \frac{\sum_{\varsigma,\tau}\binom{N_{\varsigma\tau}}{2} - \left[\sum_{\varsigma}\binom{N_{\varsigma}}{2}\sum_{\tau}\binom{N_{\tau}}{2}\right]/\binom{N}{2}}{\frac{1}{2}\left[\sum_{\varsigma}\binom{N_{\varsigma}}{2} + \sum_{\tau}\binom{N_{\tau}}{2}\right] - \left[\sum_{\varsigma}\binom{N_{\varsigma}}{2}\sum_{\tau}\binom{N_{\tau}}{2}\right]/\binom{N}{2}},$$

where $N$ is the total number of cells, and $N_{\varsigma\tau}$ represents the number of cells that are shared by the predicted cluster $\varsigma$ and true label $\tau$, $N_{\varsigma}$ and $N_{\tau}$ are the number of cells in the predicted

cluster $\varsigma$ and true label $\tau$, respectively. NMI is defined as

$$NMI = \frac{\sum_{\varsigma,\tau} P_{\varsigma\tau} \log \frac{P_{\varsigma\tau}}{P_{\varsigma} P_{\tau}}}{(-\sum_{\varsigma} P_{\varsigma} \log P_{\varsigma} - \sum_{\tau} P_{\tau} \log P_{\tau})/2},$$

where $P_{\varsigma\tau} = \frac{N_{\varsigma\tau}}{N}, p_{\varsigma} = \frac{N_{\varsigma}}{N}$ and $P_{\tau} = \frac{N_{\tau}}{N}$. Higher ARI or NMI indicates better clustering result.

## Supporting information

**S1 Fig. Clustering performance of Secuer and U-SPEC.** (A) The differences between Secuer and U-SPEC. Here the number of clusters $K$ in Secuer is estimated from data (i.e., data-adaptive) and in U-SPEC is user-specified (i.e., not data-adaptive). (B) ARI (left) and NMI (right) of Secuer and U-SPEC on 128 datasets from Mouse Cell Atlas, where Secuer used a locally scaled Gaussian kernel and U-SPEC used a non-locally scaled Gaussian kernel. The detailed information on these datasets is provided in S2 Table. Each point is the average over 10 runs and the dashed rectangles refer to the datasets with poor results (defined as those with ARI < 0.1). P-values are computed from paired Wilcoxon test. (C) Barplots of ARI (left) and NMI (right) compare the performance of Secuer and U-SPEC in those datasets with poor results identified in (B). (D) ARI (left) and NMI (right) of Secuer-consensus (i.e., Secuer-C) and U-SENC. Here U-SENC is the consensus clustering method based on U-SPEC. (E) The UMAP of clustering results by Secuer (left) and U-SPEC (right) on the Adult bladder dataset. (F) Heatmap showing the eigenvectors of the bipartite graph Laplacian of Secuer and U-SPEC on the Adult bladder dataset, where rows represent cells and columns represent the eigenvectors corresponding to the top 3 largest eigenvalues. The ground-truth labels are plotted as Group.
(TIF)

**S2 Fig. The Secuer parameters benchmarked in six datasets.** (A) The NMIs for six datasets are computed over different top numbers of principal components (pc) and different numbers of nearest neighbors in MAKNN. (B) The NMIs for six datasets are computed over different numbers of principal components and different numbers of anchors. Different panels represent different numbers of principal components.
(TIF)

**S3 Fig. Performance of Secuer and other methods on simulated datasets.** (A) The NMI of different methods on simulated datasets with different sample sizes. The simulated datasets with an increasing number of cells ranging from 10,000 to 40 million are generated from Mouse brain datasets (see Materials and Methods for more details). (B) The number of clusters estimated by Louvain in five simulated datasets with sample sizes ranging from 5 million to 9 million under different resolutions (x-axis). (C) We divided the entire clustering procedure into three steps and showed the runtime of each step taken by Secuer and vanilla spectral clustering (VSC) on four datasets, including Worm neuron, Simulation data with 10,000 samples, Mouse retina and TAM FACS with the number of cells ranging from 4,217 to 110,823. (D) The NMI of two methods on the four datasets.
(TIF)

**S4 Fig. The performance of the different methods by NMI metrics on all 15 datasets.** (A-C) The NMI of the different methods on the three large datasets (A), six gold standard datasets (B) and six silver standard datasets (C), where the COVID19-MT refers to the major cell types label, and COVID19-CT refers to the cell types label provided by author in COVID19 dataset.

(D) The summary of NMI of the different methods on 15 datasets.
(TIF)

**S5 Fig. The influence of the data preprocessing steps on clustering accuracy.** (A) Clustering performance with and without preprocessing, in which "ALL" refers to using all four preprocessing steps including normalization, logarithmic transformation, selection of high variable genes (HVG) and scaling (zero mean and equal variance), and 'None' refers to omitting all four steps. (B-E) Comparison of clustering accuracy between removing one of the steps and 'ALL', including removing normalization (B) logarithmic transformation (C), selection of high variable genes (D) and scaling (E).
(TIF)

**S6 Fig. The accuracy of the number of clusters estimated by different methods on the twelve gold/silver standard datasets.** (A-D) The Pearson correlation between the estimated number of clusters and the ground-truth (Reference) across different methods: Secuer: based on the community detection on the anchor graph (A), Secuer-eigen: based on the eigenvalues of bipartite graph Laplacian between cells and anchors (B) (see Materials and Methods), Louvain (C), and Leiden (D). The mean absolute error (MSE) for each method is shown in the top left corner of the plot.
(TIF)

**S7 Fig. The Secuer-consensus parameters benchmarked on twelve datasets.** (A-C) Clustering accuracy quantified by ARI (A) and NMI (B) vs. the number of repetitions in consensus clustering ($M$) outputs of Secuer fed into Secuer-consensus (i.e., Secuer-C) over different datasets. (C) Runtime vs. $M$ over different datasets.
(TIF)

**S8 Fig. The runtime of Secuer and Secuer-consensus using parallel computation.** (A) Clustering time of Secuer (A) and Secuer-consensus (i.e., Secuer-C) (B) vs. the number of cores used in parallel computation on different datasets.
(TIF)

**S1 Table. Overview of scRNA-seq benchmark datasets in this study.**
(XLSX)

**S2 Table. Overview of scRNA-seq benchmark datasets in S1 Fig.**
(XLSX)

## Acknowledgments

We would like to thank Dr. Franziska Michor and her group at Harvard University and Dana-Farber Cancer Institute for helpful discussions and suggestions. We would like to thank Wu and Zheng labs for helpful discussions. We gratefully acknowledge the High-performance Computing Platform of Peking University for conducting the scRNA-seq data analyses.

## Author Contributions

**Conceptualization:** Xiaoqi Zheng, Hua-Jun Wu.

**Data curation:** Nana Wei, Yating Nie.

**Formal analysis:** Nana Wei, Yating Nie.

**Funding acquisition:** Hua-Jun Wu.

**Investigation:** Xiaoqi Zheng, Hua-Jun Wu.

**Methodology:** Nana Wei.

**Resources:** Lin Liu, Xiaoqi Zheng, Hua-Jun Wu.

**Software:** Nana Wei.

**Supervision:** Lin Liu, Xiaoqi Zheng, Hua-Jun Wu.

**Validation:** Yating Nie, Lin Liu.

**Visualization:** Nana Wei, Yating Nie.

**Writing – original draft:** Nana Wei, Lin Liu, Xiaoqi Zheng, Hua-Jun Wu.

**Writing – review & editing:** Nana Wei, Yating Nie, Lin Liu, Xiaoqi Zheng, Hua-Jun Wu.

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
