## [Decision Letter · Decision Letter 0]

28 Oct 2022

Dear Dr. Zheng,

Thank you very much for submitting your manuscript "Secuer: ultrafast, scalable and accurate clustering of single-cell RNA-seq data" for consideration at PLOS Computational Biology. As with all papers reviewed by the journal, your manuscript was reviewed by members of the editorial board and by several independent reviewers. The reviewers appreciated the attention to an important topic. Based on the reviews, we are likely to accept this manuscript for publication, providing that you modify the manuscript according to the review recommendations.

Sincerely,

Piero Fariselli

Academic Editor

PLOS Computational Biology

Lucy Houghton

Staff

PLOS Computational Biology

Reviewer's Responses to Questions

**Comments to the Authors:**

Reviewer #1: This paper presents a bipartite graph based clustering algorithm for single-cell RNA-seq clustering. This work is similar to the U-SPEC algorithm in [23]. Please provide more detailed and specific discussions about the similarity and dissimilarity between the proposed algorithm and the U-SPEC algorithm. Experimental comparison between the proposed algorithm and the U-SPEC algorithm is also suggested.

In the experiments, the benchmark datasets and the baseline algorithms should be described with more details. An additional table of their statistics can help improve the clarity.

In the reference [23], multiple U-SPEC clusterers are jointly modeled via an ensemble clustering strategy. Similarly, some ensemble clustering techniques, such as multidiversified ensemble clustering (MDEC) and enhanced ensemble clustering via fast propagation of cluster-wise similarities, can also be considered in the future extensions.

Reviewer #2: The authors propose a sound and elegant clustering approach that shows notable (time and memory) efficiency for massive data while remaining accuracy-wise competitive with state-of-the-art alternatives. To this end, Secuer relies on simplistic yet highly effective principles: pivoting of anchors for scaling up efficiency, anchor neighborhood per observation, locally-scaled Gaussian kernel on the neighborhoods to better capture observation-to-anchor similarity, and automated estimation of the number of clusters.

The manuscript is written with outmost clarity, simplicity, and rigor.

The experiments are comprehensive and appropriate to assess the bold claims. Comparison against state-of-the-art approaches, as well sensitivity analysis on the parameters, is conducted across a large base of dataset, including diverse scRNA-seq datasets, complementary large real-world datasets, well-annotated benchmarks, and synthetic data.

The software is provided as an open-source module in GitHub, ensuring the reproducibility of the acquired results.

Some minor-moderate concerns:

1) consider moderating a few claims:

- "Secuer enjoys reduced runtime and memory usage by orders of magnitude" -> over one order of magnitude for datasets with more than 1 million cells

- "again greatly improves"

2) it is not always clear what is the default estimator to approximate the number of clusters in Secuer,

please clarify whether you opted to use the Louvain estimator on the anchors or the near-zero eigenvalue method. It is also not completely clear whether the later method was assessed.

In addition, consider disclosing in the overview the selected community detection method, as well as adding reference [11] after "near-zero eigenvalues of the graph Laplacian (see Materials and Methods)".

3) to fully attest the scalable nature of the proposed approach, consider further assessing whether Secuer can be parallelized/distributed and, if so, introducing high-level principles to this end

4) the intuition for the locally-scaled Gaussian kernel is a simple one, please briefly provide it in the Overview instead of referencing "Materials and Methods"

5) "The preprocessing involves four steps: 1) gene/cell filtering; 2) normalization; 3) selection of highly variable genes; 4) dimension reduction by PCA" -> please ensure that the applied preprocessing parameters are available for all the tested datasets as they critically impact experiments

6) minor notes on complexity majorants:

- (2) -> please introduce d and N right after or at the beginning ('Let d and N be...')

- (1/2) -> provide the intuition for the squared root of 'p' in the Materials and further disclose the meaning of 'o' parameter

7) proof-check for minor language aspects:

- method Secuer for -> method, Secuer, for

- Fig1F -> Fig 1F

- "It appears that" -> consider instead referencing memory 'estimates' to avoid 'appears'

- "divide the graph into disjoint subgraphs" -> please validate the use of 'disjoint'

8) few notes on the supplementary material

- reformat S1

- no information is provided on the simulated datasets in S3, please clarify the generation procedure

- consider starting from a lower number of solutions in S7 to better assess the impact of consensus

I hope these comments are useful and wish to see your work published in the near future,

Kind regards

**Have the authors made all data and (if applicable) computational code underlying the findings in their manuscript fully available?**

Reviewer #1: None

Reviewer #2: Yes

PLOS authors have the option to publish the peer review history of their article (what does this mean?). If published, this will include your full peer review and any attached files.

Reviewer #1: No

Reviewer #2: No

Figure Files:

Data Requirements:

Reproducibility:

References:

---

## [Decision Letter · Decision Letter 1]

22 Nov 2022

Dear Dr. Zheng,

We are pleased to inform you that your manuscript 'Secuer: ultrafast, scalable and accurate clustering of single-cell RNA-seq data' has been provisionally accepted for publication in PLOS Computational Biology.

Best regards,

Piero Fariselli

Academic Editor

PLOS Computational Biology

Lucy Houghton

Staff

PLOS Computational Biology

Reviewer's Responses to Questions

**Comments to the Authors:**

Reviewer #1: The authors have well addressed my concerns.

Reviewer #2: Thank you for the undertaken care and effort on revising the manuscript in accordance with the few suggestions,

The authors have successfully addressed my concerns

**Have the authors made all data and (if applicable) computational code underlying the findings in their manuscript fully available?**

Reviewer #1: None

Reviewer #2: None

PLOS authors have the option to publish the peer review history of their article (what does this mean?). If published, this will include your full peer review and any attached files.

Reviewer #1: No

Reviewer #2: No

---

## [Editor Report · Acceptance letter]

30 Nov 2022

PCOMPBIOL-D-22-01465R1 

Secuer: ultrafast, scalable and accurate clustering of single-cell RNA-seq data

Dear Dr Zheng,

I am pleased to inform you that your manuscript has been formally accepted for publication in PLOS Computational Biology. Your manuscript is now with our production department and you will be notified of the publication date in due course.

With kind regards,

Zsofi Zombor
